# Changes in Digestive Microbiota, Rumen Fermentations and Oxidative Stress around Parturition Are Alleviated by Live Yeast Feed Supplementation to Gestating Ewes

**DOI:** 10.3390/jof7060447

**Published:** 2021-06-04

**Authors:** Lysiane Dunière, Damien Esparteiro, Yacine Lebbaoui, Philippe Ruiz, Mickael Bernard, Agnès Thomas, Denys Durand, Evelyne Forano, Frédérique Chaucheyras-Durand

**Affiliations:** 1Lallemand SAS, 19 rue des Briquetiers, BP 59, CEDEX, 31702 Blagnac, France; lduniere@lallemand.com (L.D.); esparteiro.d@gmail.com (D.E.); ylebbaoui@lallemand.com (Y.L.); 2UMR (Unité Mixte de Recherches) 454 MEDIS (Microbiologie Environnement Digestif et Santé), INRAE, Université Clermont Auvergne, 63122 Saint-Genès Champanelle, France; philippe.ruiz@inrae.fr (P.R.); evelyne.forano@inrae.fr (E.F.); 3UE (Unité Expérimentale) 1414 Herbipôle, INRAE (Institut National de Recherche pour l’Agriculture, l’Alimentation et l’Environnement), Université Clermont Auvergne, 63122 Saint-Genès Champanelle, France; mickael.bernard@inrae.fr; 4UMR (Unité Mixte de Recherches) 1213 Herbivores, INRAE (Institut National de Recherche pour l’Agriculture, l’Alimentation et l’Environnement), Université Clermont Auvergne, 63122 Saint-Genès Champanelle, France; agnes.thomas@inrae.fr (A.T.); denis.durand@inrae.fr (D.D.)

**Keywords:** gestating ewes, parturition stress, rumen microbiota, fecal microbiota, live yeast, DNA sequencing, oxidative stress

## Abstract

Background: In ruminants, physiological and nutritional changes occur peripartum. We investigated if gastro-intestinal microbiota, rumen metabolism and antioxidant status were affected around parturition and what could be the impact of a daily supplementation of a live yeast additive in late gestating ewes. Methods: Rumen, feces and blood samples were collected from 2 groups of 14 ewes one month and a few days before parturition, and 2 weeks postpartum. Results: In the control ewes close to parturition, slight changes in the ruminal microbiota were observed, with a decrease in the concentration *F. succinogenes* and in the relative abundance of the Fibrobacteres phylum. Moreover, a decrease in the alpha-diversity of the bacterial community and a reduced relative abundance of the Fibrobacteres phylum were observed in their feces. Control ewes were prone to oxidative stress, as shown by an increase in malondialdehyde (MDA) concentration, a lower total antioxidant status, and higher glutathione peroxidase (GPx) activity in the blood. In the yeast supplemented ewes, most of the microbial changes observed in the control group were alleviated. An increase in GPx activity, and a significant decrease in MDA concentration were measured. Conclusions: The live yeast used in this study could stabilize gastro-intestinal microbiota and reduce oxidative stress close to parturition.

## 1. Introduction

In dairy ruminant systems, the period around parturition is characterized by important hormonal, physiological, psychological, and nutritional changes and thus imposes severe challenges for the animal. Animals may develop metabolic stress if they fail to physiologically adapt to these challenges. The Negative Energy Balance (NEB) commonly encountered by the ruminant peripartum results in increased lipid mobilization and the generation of Reactive Oxygen Species (ROS). This alters the immune and metabolic status and ultimately leads to an increased risk of metabolic and infectious diseases such as enterotoxemia, ketosis, or mastitis during the transition period [1]. The dam also has a primary role in passive immune transfer through the colostrum [2] and in digestive microbiota inoculation [3] to the newborn.

At the gastrointestinal (GIT) level, distinct prepartum to postpartum shifts in the rumen bacterial community composition have been observed in cows between 3 weeks before and 3 weeks after calving, explained at least in part by the change in diet composition, notably by the increase in the proportion of readily fermentable carbohydrates, which is necessary to ensure the energy needs of the cow around parturition [4,5,6]. Such changes may put the animal at risk for rumen dysbiosis, leading, for instance, to acidosis, and leading to alterations in the rumen epithelium function [7]. Moreover, it has been shown that the microbial components from Gram negative bacteria, such as lipopolysaccharides (LPS), can be released in the disturbed rumen and can trigger a proinflammatory response as well as the production of reactive oxygen species [8,9]. In ewes, in which severe feed restriction in pregnancy may be observed, most ruminal fermentation parameters have been shown to decrease, associated with changes in the composition of the rumen epithelium-associated microbiota, and deleterious effects on the barrier function of the rumen epithelium have been reported [10].

For all these reasons, an optimal preparation of gestating ewes the few weeks before parturition is crucial to ensure the parturition process and the beginning of lactation in optimal conditions. Nutritional strategies that are able to prevent the development of oxidative processes during pregnancy [11], and an optimal management of the rumen function appear as key elements that participate in the maintenance of a good nutritional and health status in the gestating ewe. Several reports have demonstrated the benefits of live yeasts on rumen pH stabilization, microbial populations, and fiber digestibility [12,13,14] in sheep or cow studies.

In this work, we characterized the changes occurring around parturition on the rumen and fecal microbiota abundance, diversity and taxonomic composition, rumen fermentations, and oxidative stress. In addition, we studied the impact of live yeast (*Saccharomyces cerevisiae* CNCM I-1077, Levucell SC, Lallemand Animal Nutrition, Blagnac, France) supplementation prepartum with the hypothesis that this would represent an effective strategy to improve the rumen environment, prevent rumen dysbiosis, and protect animals against oxidative stress.

## 2. Materials and Methods

### 2.1. Diets and Animals

Twenty-eight gestating ewes (*Ovis aries*, Romane breed) were used for this study. All of them carried two or three fetuses. They were selected following an echographic evaluation a few weeks before the start of the trial and were assigned to two groups (control = C, supplemented = SC) which were balanced homogeneously according to age, parity, body condition score, and live weight.

One month and a half before the estimated date of parturition, the ewes were transferred from the farm unit to a room equipped with Biocontrol CFRI systems (Biocontrol, Rakkestad, Norway) which allowed for the control of the concentrate intake, to ensure that ewes received the full yeast treatment. Indeed, because the yeast supplement was incorporated into the experimental concentrate, it was important to ensure that each animal had the same quantity of concentrate ingested, and at the same time of the day. The ewes were identified by means of ear RFID (radio frequency identification) transponders for specific access to the manger. The ewes were progressively adapted to the concentrate during the month before the start of the trial, by increasing the amount of concentrate fed daily up to 800 g/d/animal. Then, until parturition, each ewe received this fixed amount of concentrate (Moulin de Massagettes, Massagettes, France, Appendix A) daily, which was distributed once at 8:00 am, and which was followed by 2 kg of meadow hay. Good quality water was offered ad libitum. The nutritional composition of the diet is detailed in the Appendix A.

After 2 weeks of adaptation to the BioControl system and to this diet, the two groups received their allocated experimental concentrate, the only difference being the incorporation of the live yeast product in the supplemented group. The live yeast product *Saccharomyces cerevisiae* CNCM I-1077 (Levucell SC TITAN, Lallemand SAS, Blagnac, France) was a coated formulation included with the concentrate ingredients during the pelletization process, which allows for the protection of live yeast cells during the concentrate production. The rate of inclusion in the concentrate was calculated to bring 8 × 10^9^ CFU per day per individual. A few days before their estimated date of parturition, the ewes were transferred to a maternity unit which was split into two large pens separated by a concrete wall, to ensure that no contact could occur between the groups. The bedding was made of straw. The animals were then kept in these pens until the end of the experiment. The weight of the offspring was recorded within the first hour following the birth.

After parturition, the concentrate was not supplemented anymore with the live yeast product. The composition of the postpartum diet was modified to meet the requirements of the dam in order to feed only one lamb (the other one was directed to an artificial milk feeding system). So, each ewe was fed with 600 g of concentrate and 3 kg of meadow hay, covering a bit more than 135% of its energy needs.

The total yeast enumeration was performed in experimental concentrates throughout the study (the day of concentrate delivery, and 15, 30, and 60 days after) to ensure that the concentration of yeast met the expectations. Briefly, 30 g of pellets were ground for 30 sec in a Waring blender, then suspended in peptone water, ground again for 1 min in the Waring blender, transferred in a stomacher, and homogenized for 1 min in a sterile filter bag. One milliliter was then collected from the bag and diluted in 9 mL of sterile peptone water. Serial dilutions were performed and plated onto Sabouraud + Chloramphenicol agar Petri dishes (AES Chemunex/BioMérieux, Combourg, France). Duplicate plates per dilution were incubated during 48 h at 30 °C before colonies could be counted. In the control concentrate, no viable yeast was detected on the agar plates. In the supplemented concentrate, the average concentration from the duplicate analysis was 1.03 × 10^7^ CFU per gram of feed, so the concentration matched the expectations (i.e., with 800 g fed daily, to bring 8 × 10^9^ CFU per head).

The animal trial was conducted at the animal facilities of INRAE Herbipôle Experimental Unit UE1414 (Clermont Auvergne Rhône Alpes, Saint-Genès Champanelle, France). The procedures on the animals were carried out in accordance with the guidelines for animal research of the French Ministry of Agriculture and all other applicable national and European guidelines and regulations for experimentation with animals (see https://www.legifrance.gouv.fr/loda/id/JORFTEXT000027038013/ for details, accessed on 3 June 2021).

The protocol was favorably evaluated by the Regional Ethics Committee for Animal Experimentation C2EA-02 and the French Research Ministry authorized its implementation with the reference number 14981-2018050417167566V3 (21 December 2018).

### 2.2. Sample Collection

All samples were collected at different periods: the week before the start of the supplemented concentrate distribution i.e., three to four weeks before parturition (BS for ‘before live yeast supplementation’), a few days (8 on average) before parturition (Pa for ‘close to parturition’), and two weeks after parturition (PP for ‘postpartum’). The rumen fluid (~100 mL) was collected the same day for all the ewes before the morning feeding via a stomach PVC tube which was connected to a manual pump. The retention of the ewes was smooth, and the tube was carefully introduced in the mouth and pushed gently inside the rumen. The regurgitated digestive contents were retrieved in a sterile container. The quality of the sample was visually checked (absence of a visible amount of saliva, no trace of blood). Immediately after sampling, the pH was recorded, and the samples were brought back to the laboratory where they were processed. One portion was treated for analysis of volatile fatty acids (VFA), the other was frozen at −20 °C for microbial analysis with molecular methods. The fecal samples (~50 g) were obtained by rectal collection and rapidly frozen at −20 °C for further analysis.

Whole blood samples were collected two days after the collection of the digestive samples via venipuncture in the jugular vein, and plasma samples were prepared for further oxidative stress biomarkers, as previously described [15], and stored in light safe Eppendorf tubes at −80 °C until analysis.

### 2.3. Measured Parameters

The rumen pH was measured immediately after sampling using a laboratory pH probe. VFA analysis was performed on the rumen samples as previously described [16]. The rumen fluid was centrifuged for 10 min at 10,000× *g*, 4 °C. DNA was extracted from 250 mg of rumen centrifuged pellets or from 150 mg of fecal contents using the Quick DNA Fecal/Soil Microbe kit (Zymo Research, Irvine, CA, USA). DNA yield and quality were determined after Nanodrop 1000 spectrophotometric quantification. The DNA extracts were stored at −20 °C until analysis.

The microbial populations were quantified using the qPCR method, with specific primer sets and PCR conditions targeting the ribosomal RNA genes of total bacteria, protozoa, archaea, the yeast *S. cerevisiae,* specific bacterial groups, genus, or species according to the digestive sample (rumen fluid or feces) and the Internal transcribed spacer 1 (ITS1) of rumen fungi. The PCR targets and primers are summarized in the Appendix A.

Standards were used to determine the absolute abundance of the microbial groups, expressed as the Log_10_ number of gene copies per microgram of the pelleted rumen or feces. For total bacteria, cellulolytic bacteria, and methanogenic archaea, the standard curves were prepared according to Mosoni et al. [17]. For protozoa, the standard curve was prepared according to Bayat et al. [18]. For each target, a standard curve was prepared from 10^2^ to 10^9^ copies by serial dilution. For *S. cerevisiae* quantification, the standard curve was constructed using DNA obtained from Levucell SC20 commercial product. A PBS-suspension of 10^8^ CFU/g was prepared (total CFU/g in the commercial product was previously checked) and DNA extraction was performed on this suspension. Decimal dilutions of DNA were performed to obtain a range of corresponding cell concentrations of 10^7^ to 10^3^ CFU/g. It was then possible to obtain a standard curve relating the yeast cell concentrations and the Cycle threshold (Ct) values. The efficiency of the qPCR method for each target varied between 97 and 102% with a slope from −3.0 to −3.4 and a regression coefficient above 0.95.

Microbiota diversity and taxonomic composition were analyzed by a 16S/18S amplicon metagenomic sequencing approach. DNA samples were quantified with a Qubit spectrophotometer to adjust the concentrations to at least 20 ng/µL, and a volume of 30 µL per sample was sent to the Novogene sequencing platform (Novogene Co. Ltd., Cambridge, UK). DNA sequencing was performed on a subset of samples from 6 ewes per group balanced in terms of age, weight, and body condition score, for each sampling time considered. The diversity and composition of rumen/fecal microbiota were studied using the high throughput sequencing Illumina MiSeq (Illumina, San Diego, CA, USA) method (2 × 250 nt paired ends). The primer sets used, and the rDNA regions targeted are indicated in the Appendix A. Libraries construction and MiSeq Illumina sequencing were carried out following the protocols validated by Novogene. The paired-end reads were merged and filtered, and the chimera were removed using FLASH, the Qiime quality control process and the UCHIME algorithm, respectively [19,20,21,22]. A sequence analysis was performed by the Uparse software with ≥97% similarity threshold [23,24]. For each OTU representative sequence, Mothur software was performed against the SSUrRNA database of the SILVA Database (Threshold:0.8 ~ 1) [25,26]. Subsequent analyses of alpha diversity and beta diversity were performed with R studio software using Phyloseq and Microbiome R packages [27,28] on normalized samples. The differential abundance analysis at OTU level was performed with the DESeq2 package [29].

The measurement of the ferric reducing ability of plasma (FRAP) was completed by the assay based on the method of Benzie and Strain [30] but was slightly modified. Briefly, 30 μL of sample and 90 μL of water were pipetted in the microplate in triplicate. A working reagent (150 μL at 10:1:1 of acetate buffer 300 mmol/L pH 3.6: FeCl_3_ solution 20 mmol/l:2,4,6-tripyridyl-s-triazine solution 10 mmol/L) was added in each well, and the reaction mixture was incubated for 30 min at 37 °C. The absorbance was measured at 593 nm. The GPx activity was measured spectrophotometrically according to Agergaard and Jensen [31].

The indicators of energy and lipid metabolism were measured by spectrophotometry as indicated by Delosière et al. [32]. The measurement of malondialdehyde (MDA) was determined in the plasma samples by high performance liquid chromatography (HPLC) followed by UV spectrophotometry [33].

The indicators of liver status (IU/L) were measured directly from plasmatic samples by spectrophotometry using specific kits according to the manufacturers’ recommendations, as described in Delosière et al. [32]. The aminotransferases were measured with a Sobioda kit (ALAT, Montbonnot-Saint-Martin, France) and a Biodirect TGO kit (ASAT, Lavilleneuve, France). The phosphatase alkaline was measured with a ThermoFisher Scientific kit (PAL, Waltham, MA, USA), and gamma-glutamyl transpeptidase with a Biodirect kit (GGT, Lavilleneuve, France).

### 2.4. Statistical Analyses

Graphical representations and statistical analyses were performed using GraphPad Prism v8.4.3. The results are presented as the mean ± SD. The animals were assigned to C or SC groups according to age, parity, body condition score, and live weight. However, significant differences (*p* < 0.05) or tendency (*p* < 0.10) were observed BS for several parameters (i.e., pH, Acetate, Propionate, and relative abundance of Euryarchaeota, *Entodinium* and total anaerobic fungi in rumen; alpha diversity indices and relative abundance of Bacteroidetes in feces; MDA and FRAP in blood). Consequently, a new variable Y_diff_ was defined for each parameter considered. Y_diff_ corresponded to the subtraction of the initial BS values to the Pa and the PP values. A linear mixed model with repetitions was then applied to these new Y_diff_ variables to evaluate the effect of group and time factors and their interaction. This “anova of change” model has been previously recommended by Van Breukelen [34] in order to limit the bias observed in the case of preexisting groups presenting significantly different baselines. For some parameters, the data could not fit the chosen mixed model i.e., when the data were not normally distributed and the non-parametric Mann–Whitney (MW) test was thus applied to compare the C and SC groups at each time. Therefore, statistical analyses are presented either in tables summarizing the *p*-values from the linear mixed model, or in tables summarizing the *p*-values from the Mann–Whitney tests in the case of data that were not normally distributed. For the oxidative parameters, the samplings were performed only at BS and Pa, and an unpaired *t*-test without assuming an equal SD (Welch test) was performed on the normalized Pa values. The statistical significance was determined at a *p*-value < 0.05 and trends were discussed when *p* < 0.10. To ease comprehension, graphical representations of the Y_diff_ variables are presented in the Appendix A section only for the parameters with *p*-values ≤ 0.10.

## 3. Results

### 3.1. Rumen pH and VFAs

The mean rumen pH, which was measured before the morning feeding, was quite stable throughout the whole experimental period (Figure 1A). The pH values were close to neutrality or even higher, maybe due to small saliva contamination during the rumen sampling through the oral tubing. No effect of either the period or the treatment was observed on the rumen pH (*p* > 0.05).

The total VFA, acetate, propionate, and butyrate concentrations presented different evolutions between the control and SC groups (Figure 1B). Indeed, the total VFA and acetate concentrations decreased at Pa and increased at PP in the control group while they remained stable in the SC group (Table 1, Appendix A). A significant time effect was observed for Valerate and Caproate with a tendency of decrease PP for Valerate.

### 3.2. Rumen and Feces Microbiota

#### 3.2.1. qPCR Results

In the rumen, *Saccharomyces cerevisiae* was detected in both groups before supplementation (5.88 ± 0.49 and 5.65 ± 0.37 Log_10_ copies/g of pelleted rumen content for the control and SC groups, respectively, Table 2) in the SC group. A strong statistical effect of both time and group factors was observed as the abundance of this species increased significantly in the SC group in the samples collected just before parturition (Appendix A): it went up to 7.49 ± 0.34 Log_10_ copies/g of pelleted rumen content, whereas it remained stable in the control group (6.11 ± 0.60 Log_10_ copies/g of pelleted rumen content). Two weeks after parturition, the concentrations of *Saccharomyces cerevisiae* decreased in the SC group to values comparable with the control group, and to values quite similar to those found at the start of the trial.

The total bacteria concentration was little affected over time as it very slightly increased (*p* < 0.05) just before parturition in both groups and decreased postpartum (Table 2, Appendix A). The concentration of *Prevotella sp*. decreased postpartum, whatever the group (*p* < 0.05). *Fibrobacter succinogenes* was much more abundant than the two *Ruminococcus* species, with concentrations above 9 Log_10_ 16S copies/g of pelleted rumen content and represented ~5% of the total bacteria population. The significant difference in the *R. flavefaciens* concentration in the SC group compared to the control group was mainly due to a high BS average while *R. albus* was increasing over time in both groups. In the control group, *F. succinogenes* abundance decreased by 53.2% from BS to Pa, then slightly increased at PP, while it decreased by only 18.7% from BS to Pa in the SC group to finally reach 74.1% of its initial abundance at PP (Table 2). However, these differences were not found to be significant (*p* = 0.999 and *p* = 0.523 at Pa and PP respectively, Mann–Whitney test between the C and SC groups).

In the fecal samples, the abundance of *Saccharomyces cerevisiae* was quite similar to that found in the rumen (Appendix A) with a significant effect of time (*p* < 0.001), group (*p* = 0.016), and interaction (*p* = 0.002) according to the linear mixed model. A higher concentration of *S. cerevisiae* was observed in the SC samples collected just before parturition. The Q-PCR results on the other targets in the feces are presented in the Appendix A. They did not differ according to the time and the group or their interaction (*p* > 0.05).

The concentration of methanogenic Archaea was 8 Log_10_ *mcrA* gene copy numbers/g, which was similar to that found in the rumen. *Escherichia coli* was quantified between 8 and 9 Log_10_ 16S gene copy numbers/g, which represented 0.13% of the total bacteria concentration. *Faecalibacterium prausnitzii* was quantified at the same level. Regarding the fibrolytic microorganisms, anaerobic fungi were detected at less than 5 Log_10_ ITS copies/g, so 1 to 2 Log_10_ lower than the concentrations found in the rumen, and *Fibrobacter succinogenes* was weakly detected at a level close to 6 Log_10_ 16S gene copies/g, representing 0.001% of the total bacterial population.

#### 3.2.2. 16S-DNA Sequencing Results

Alpha and beta diversity measures

A great variability between individuals was noticed in the alpha diversity measures of the rumen bacterial communities whatever the group and physiological stage (Appendix A). No significant difference was measured overall between the control and the SC groups (Table 3). However, there was a significant time effect, indicating a decrease in the richness and evenness at Pa followed by an increase at PP. No group effect was observed for the InvSimpson index (*p* > 0.05, Mann–Whitney test on normalized values).

Less variability between the individuals was observed in the fecal diversities compared to the ruminal diversities (Appendix A). A significant decrease in richness and evenness was noticed in the Control group at parturition, compared to the SC group (Table 4).

The beta diversity was also studied for both the ruminal and fecal communities using the Bray-Curtis dissimilarity matrix (Figure 2). The ruminal communities were rather close to each other between the individuals for a given physiological stage. No treatment effect could be observed. The fecal microbial community structure from the control group at BS appeared very different from the ones analyzed afterwards (Permanova analysis, *p* < 0.05). In the SC group, the structure was more stable.

Relative abundances of the main taxonomic groups

Almost 98% of the 16S sequences were assigned to nine main phyla (Figure 3). Among these nine phyla, Bacteroidetes was largely dominant, followed by Firmicutes and Fibrobacteres.

A significant decrease in the Bacteroidetes phylum was observed in the ruminal contents only after parturition (PP) while a tendency for an increase in Actinobacteria was observed (Table 5, Appendix A). Significant time effects were observed on Proteobacteria and Spirochaetes which were slightly increased at postpartum in both groups. A lower relative abundance of Euryarchaeota was observed in the SC group at Pa, indicating a decrease during the supplementation period. The relative abundance of Verrucomicrobia was lower in the SC group postpartum (*p* = 0.051) according to the Mann–Whitney test on normalized values. Although non-significant, a stronger decrease in Fibrobacteres was observed from BS to Pa in the control group (−38%, from 7.9% to 4.9%) than in the SC group (−8%, from 6.3% to 5.8%).

In the fecal samples, more than 99% of the sequences were affiliated to the same nine main phyla compared to those found in the rumen. However, Firmicutes were largely dominant, with more than 55% of the total sequences overall. Bacteroidetes represented between 27.4 and 34.2% of the sequences on average (Figure 4).

In the feces, a different evolution in the relative abundances of Actinobacteria (mainly Coriobacteriaceae and Bifidobacteriaceae families) was observed over the experiment between the groups (Appendix A). In the control group, the relative abundance of Actinobacteria was continuously decreasing from 0.87% to 0.32%, while there was a higher relative abundance in the SC group at Pa and PP (*p* = 0.015 and *p* = 0.009 for Pa and PP, respectively, according to the Mann–Whitney test on normalized values). Bacteroidetes decreased at Pa in both groups but increased to higher values at PP in the SC group compared to the control group (Table 6), and this evolution was partly linked to the relative abundance of Prevotellaceae. The relative abundance of Fibrobacteres significantly decreased at Pa, with more drastic variations observed in the control group (−76%, from 1.6% to 0.4%) compared to the SC group (−17%, from 0.7% to 0.5%). An increase in Fibrobacteres was observed in both groups at PP. A time effect was observed for Proteobacteria which were decreasing at Pa and increasing afterwards.

Differential analysis of OTUs
A DESeq2 analysis was performed on the 16S data at the OTU level. The analysis confirmed a significant difference between the two groups before supplementation.

In the rumen, it should be noted that a sample from one ewe in the Control group was highly enriched in *Listeria* just before parturition (22 Log_2_ fold change C vs. SC at Pa, corresponding to 0.7% relative abundance), and postpartum, a comparable level of enrichment was found but it was in another ewe (24 Log_2_ fold change C vs. SC at PP, corresponding to 4.0% relative abundance). Six OTUs were found in higher abundance BS compared to Pa: *Ruminobacter* in the SC group and Ruminococcaceae, *Phocaeicola*, Rikenellaceae, and *Sphaerocheata* in the control group. In the rumen, *Listeria* was enriched at Pa in the control group vs. BS, as previously noted. Between Pa and PP, differentially abundant OTUs were only found in the SC group, with an enriched abundance of several Prevotallaceae OTUs at Pa compared to PP.

In the feces, several OTUs were observed in higher abundance in the C or SC groups before supplementation: 3 OTUs in the SC group and 18 OTUs in the control group. *Listeria* was more represented in the feces of the supplemented group before supplementation than at parturition (8.2 Log_2_ fold change BS vs. Pa in the SC group, average of 0.2% relative abundance). The temporal analysis was thus driven by the differences observed BS. In the feces, 29 OTUs were more represented in the samples collected before supplementation than at Pa, all in the control group. *Bifidobacterium* and *Paeniclostridium* were the only 2 OTUs enriched in the SC group at Pa (4.76 and 1.98 Log_2_ fold change Pa vs. BS in the SC group, respectively). *Fibrobacter* related OTU decreased in the control group at Pa, compared to BS. Two weeks postpartum, *Bifidobacterium* OTU was enriched in both the groups compared to just before parturition (4.97 and 4.84 Log_2_ fold change PP vs. Pa in the C and SC groups, respectively) and *Fibrobacter* related OTU was enriched in the control group.

#### 3.2.3. 18S-DNA Sequencing Results

Alpha and beta diversity measures
No significant difference in the alpha diversity indexes was observed over time and between the groups according to the Mann–Whitney test on normalized values (Appendix A, *p* > 0.05). We noticed that at Pa, the Shannon index was very variable among the animals in the control group, compared to other time points and to the SC group. The representation of beta diversity indicates that the ruminal samples from the control and SC groups at Pa clustered distinctly from other samples along the PC1 (42.4%) and PC2 (18.7%) axis and were also clearly discriminated from each other (Figure 5). No other cluster could be easily observed.

Relative abundance of the main taxonomic groups
Eukaryotic communities were studied according to separated functional groups (protozoa, total fungi, anaerobic fungi). The protozoal population was represented by 12 genera (Figure 6). Overall, the genera from the Litostomatea class strongly dominated the eukaryotic populations with a total relative abundance ranging from 99.6 to 99.8%. *Entodinium* related sequences were the most abundant, followed by *Isotricha*, *Metadinium*, and *Dasytricha*.

Most of the statistical significances for the protozoa population were linked to the time effect (Table 7, Appendix A). It was observed that the relative abundances of *Metadinium*, *Polyplastron*, and unidentified Listostomatea significantly decreased at Pa and then increased afterwards. In contrast, the relative abundances of *Dasytricha* and *Isotricha* increased at Pa then decreased at PP. No significant group effect was observed but some interactions (Time × Group) were found to be significant. A greater stability in relative abundance was observed for *Isotricha* and *Eudiplodinium* in the SC compared to the control group across time. The abundance of *Entodinium* was stable in the SC group up to Pa then slightly decreased while the opposite trend was observed in the control group (a decrease at Pa then an increase at PP). No group effect was observed for *Diploplastron* according to the Mann–Whitney test on normalized values.

Great variations of the three main phyla relative abundances (Neocallimastigomycota, Ascomycota and Basidiomycota) were observed over time (Figure 7). In both groups, parturition was associated with a significant and strong increase in Ascomycota and a concomitant decrease in the relative abundance of Neocallimastigomycota (Table 8, Appendix A). At Pa, Ascomycota were observed in a higher relative abundance in the SC group compared to the control group. This was explained by the significant variations of Saccharomycetales and more precisely by the higher increase in *Saccharomyces* (from 0.3% to 5.3% of fungal abundance) in the SC group compared to the control group at Pa (*p* < 0.001, Mann–Whitney test). The higher relative abundances of *Saccharomyces* and Saccharomycetales in the SC group remained postpartum (*p* < 0.05, Mann–Whitney test) although both groups presented a decrease from Pa to PP. The relative abundances of Basidiomycota varied differently between the groups as it increased at Pa in the control group and decreased in the SC group. Similar relative abundances were observed PP in both groups. No group effect was observed for Chytridiomycota according to the Mann–Whitney test on normalized values.

Among Saccharomycetales, a significant interaction of time and group was observed for *Pichia* (*p* = 0.0248) which was observed only at the beginning of the trial in the control group and was not retrieved afterwards, while it remained at a low relative abundance in the SC group up to Pa. No effect of the factors and their interaction was observed for *Debaryomyces*, *Candida*, and total non-*Saccharomyces* taxa (linear mixed model or Mann–Whitney test, data not shown). Anaerobic fungi related sequences were identified as belonging to *Orpinomyces*, *Cyllamyces*, and Neocallimastigaceae family members as well as Chytridiales order members (Appendix A).

A significant difference in the total anaerobic fungal population was observed over time. These variations were linked to the Neocallimastigaceae family and *Cyllamyces* abundances (Table 9). No group effect was observed for unidentified Chytridiales according to the Mann–Whitney test on normalized values.

Differential analysis of OTUs
Compared to the observations on the bacterial microbiota, fewer OTUs (3) were identified as differentially abundant between the SC and C groups. *Saccharomyces* OTUs were significantly enriched in the SC group compared to the control group at Pa and PP but also at BS. Two OTUs belonging to the Agaricomycetes order and *Dibaeis* genus were highly enriched in the control group at Pa (25.26 and 24.23 Log_2_ fold change Pa vs. BS, respectively).

Ten OTUs were identified as significantly more represented before the SC supplementation compared to Pa. *Lichtheimia* and *Mucor* were the two genera more represented BS in both the SC and C groups. *Saccharomyces* was found to be significantly enriched also in both groups a few days before parturition with similar Log_2_ fold changes (−5.59 in the control groups and −5.73 in the SC groups, representing an increase between BS and Pa in relative abundance from 0.02% to 0.7% and from 2.4% to 53.0%). This observation could also be made when comparing the period Pa and PP, with −6.84 and −7.07 Log_2_ fold changes in the SC and control groups, respectively, indicating a decrease in the relative abundance during this period. From the 19 OTUs differentially observed during the pre-parturition period (comparison BS–Pa), 11 evolved the opposite way during the post-parturition period (Pa–PP), indicating that these OTUs were coming back to their initial levels.

### 3.3. Blood Biomarkers

#### 3.3.1. Oxidative Stress Blood Markers

A decrease in the FRAP was observed for both groups close to parturition (from 0.30 to 0.28 mmol Fe^2+^/L in the control group and from 0.31 to 0.26 mmol Fe^2+^/L in the SC group, Figure 8A). This antioxidant capacity was significantly lower for the SC group (*p* = 0.012). The GPx activity increased around parturition for both groups. More precisely, a 29% increase (from 0.82 to 1.06 µmol/min/mL) in the GPx activity was measured in the SC group around parturition, however the difference was not significant (*p* = 0.199). Finally, a significant effect of group was observed for MDA (*p* = 0.003). In the control group, MDA increased from 0.14 to 0.21 µg/mg protein from BS to Pa, whereas it decreased in the SC group from 0.20 to 0.18 µg/mg protein.

#### 3.3.2. Metabolic Status Blood Markers

As parturition approached, the NEFA concentration increased in both groups (+0.42 mM and +0.34 mM in the control group and the SC group, respectively, Figure 8B) while the blood glucose concentration slightly decreased (−0.07 g/L and −0.05 g/L in the control group and the SC group, respectively). The BHBA variations were different between the two groups as it increased in the control group and decreased in the SC group. However, none of these variations were found to be significantly different (*p* > 0.05).

Liver function was also evaluated through aminotransferases (alanine and aspartate aminotransferases: ALT, AST), gamma-glutamyl transferase (γ-GT), and alkaline phosphatase (ALP) activities. No significant difference was found considering the time and group factors or their interaction for the enzymatic activities tested due to quite high intra-individual variations (*p* > 0.05). Different variations were observed for the γ-GT activity which decreased in the control group but increased in the SC group, but the variation in this latter group was driven by one animal only (the same animal being slightly higher in AST activity).

### 3.4. Lambs Birth Weight

Nine out of 14 ewes from the control group and 12 out of 14 ewes from the SC group gave birth to twin lambs and only these pairs were taken into account for birth weight analysis. The average weight of the lambs born from the control ewes was 3.74 kg ± 0.52 whereas the average weight of the lambs from the SC group was 4.01 kg ± 0.80, which represented a 7.2% increase, although this was not significant (*p* > 0.05). The remaining ewes gave birth to either one single lamb (two ewes from the control group, one for the SC group) or to three lambs (three ewes from the control group, one for the SC group).

## 4. Discussion

While recent data are available on the changes that may occur in the rumen and the fecal microbial abundance, diversity, and taxonomic composition around parturition in dairy cows [4,6,35], and on the beneficial effects which are observed in the case of live yeast supplementation during this risky period [4,7], to our knowledge data on small ruminants are very scarce, in particular on ovines. The peri-parturition period is considered very stressful for the female because many profound changes in hormonal status, physiology, and metabolism occur. Metabolic changes are partly triggered by a change in diet in dairy cows, as high levels of readily fermentable carbohydrates are fed to ensure the nutritional requirements at the start of the lactating phase. Thus, the shifts in the microbiota balance in both the rumen and the lower gut can be, at least in part, explained by the modifications of both the level and the nature of nutrients that are offered to the microorganisms. For instance, in the study of Bach et al. [4], the increase in non-fiber carbohydrates in the diet was between 6 and 7 kg/day per cow, which obviously put the cows at more risk for ruminal acidosis. Thus, in dairy cows, where the inclusion of a high fermentable concentrate is compulsory to maximize milk production after calving, the impact of the hormonal, physiological, and metabolic stresses can have very negative consequences on the performance and overall health of the animal.

In our study with ewes, the nutritional context was different, as the animals were fed a high forage diet across the peri-parturition period and the level of the rapidly fermentable concentrate was not increased after lambing, because the objective was not to maximize milk production. Therefore, the diet offered at parturition was not particularly at risk for ruminal acidosis, which we indeed did not observe throughout the trial.

### 4.1. What Was the Microbial Profile in Gestating Ewes?

One month before lambing, the composition of the rumen microbiota in the ewes was quite comparable to what has been reported in the literature in adult ovines. Bacteroidetes were dominant (50–60%), followed by Firmicutes (25–30%), Fibrobacteres (5–8%), Euryarchaeota (2–7%), Spirochaetes (2%), Verrucomicrobia (1–2%) and Proteobacteria (1–2%). We noticed a quite high abundance of *Fibrobacter*, which represented around 5–8% of the total bacterial read by 16S sequencing, and around 5% of the total bacteria quantified by qPCR. The quite high abundance of Fibrobacteres is likely due to the high forage content of the diet. The species *F. succinogenes* was clearly dominant compared to the other fibrolytic species *Ruminococcus flavefaciens* and *R. albus*, which were quantified at much lower levels in our study. These results are in agreement with the data obtained by Mosoni et al. in adult sheep [17]. The qPCR detection of *Megasphaera elsdenii*, an important lactate utilizer, was very weak whatever the physiological stage, which is in agreement with the fact that the diet was low in readily fermentable sugars which are generally responsible for massive fermentation, lactate accumulation, and can lead to acidosis [36].

Rumen protozoa were quantified at levels which agree with the literature [37,38]. The 18S sequencing data showed a dominance of the *Entodinium* genus, which is fully in accordance with previously reported data, with small Entodinomorphs accounting for 95% of the total ciliate population in sheep [39]. Among the 18S sequences attributed to fungi, the typical ruminal anaerobic taxa Neocallimastigomycota was detected at a high relative abundance (more than 65%), with assigned sequences to *Orpinomyces* and *Cyllamyces* genera. It can be suggested that the high forage diet used in this study promoted this important fibrolytic phylum [40].

### 4.2. What Happened to the Digestive Microbiota and Activity of Control Ewes Around Parturition?

In the rumen, the bacterial diversity was impacted by parturition as indicated by the significant time effect on the richness and the evenness indexes. Some changes in the abundance of certain groups were measured and microbial activity seemed to be impacted as well, as shown by the VFA pattern. Several studies have shown that the rumen microbiota balance altered around parturition in cows, which was, most of the time, tightly linked to a change in diet, with higher levels of readily fermentable carbohydrates. Indeed, DNA sequencing studies on the rumen microbiota have reported a decrease in the fibre-degrading bacterial populations such as the Fibrobacterales and Clostridiales orders, and an increase in Bacteroidales, with a particular increase in the genus *Prevotella,* after calving [41].

In our study, a significant time effect was observed for Proteobacteria and Spirochaetes in the rumen of the control ewes, with an increase in the relative abundance in those phyla over time. We noticed a numerical decrease in the *Fibrobacter succinogenes* concentration, measured by qPCR, in the rumen of the control ewes at lambing, compared to one month before parturition. This was corroborated with 16S sequencing data which showed an almost 40% decrease in the relative abundance of the Fibrobacteres phylum, which gathers only one genus, *Fibrobacter*. This important fibrolytic bacterial species *s* plays a key role in fiber digestion [42,43,44]. This species has been reported to be quite sensitive to changes in the ecological conditions of the rumen [45], which might have occurred at parturition. As *F. succinogenes* represents a high proportion of the total bacteria in the rumen of our ewes and given its high contribution to plant cell wall degradation, even a small decrease in its concentration could impact fiber digestibility. Only a very few OTUs were found to be differentially represented around parturition. A very marked result was the enrichment in *Listeria*-affiliated sequences in the rumen of the control ewes around parturition. However, this observation was linked to only one individual at 0.7% at Pa. Regarding eukaryotic communities, we observed a 0.4 Log_10_ decrease in the ciliate protozoa concentration after parturition only (*p* = 0.062 with the Wilcoxon test). Variations across the periods in the protozoa and fungi taxonomic composition were also observed: a significant decrease in the relative abundance of anaerobic fungi (*Neocallimastigomycota* and *Cyllamyces*) was measured around parturition, while Ascomycota (*Saccharomyces*) increased. Variations in relative abundances were also observed in the protozoa population. Overall, our results suggest that around parturition, changes in the ecological conditions in the rumen could occur and become less favorable to the stability of the eukaryotic community.

In the control ewes, a numerical decrease in the total VFA concentration (about −9%), mainly reflected by a decrease in acetate (about −8%), was observed a few days before parturition, followed by an important increase postpartum (about +29%). This may reflect an instability of microbial fermentations.

In the feces of the control group, a strong decrease in bacteria alpha diversity has been noticed at lambing followed by an increase after parturition. Bach et al. measured a decrease in the bacteria diversity in the colon of cows a few days after calving, lasting even 3 weeks postpartum [4]. This difference could be, at least in part, explained by the diet composition in the two studies, as in the Bach et al. study, readily fermentable carbohydrates were increased in the post calving diet, which was probably more prone to an increased risk for lower gut dysbiosis that the one used in our study. Fecal 16S sequencing data indicated that the relative abundance of the Bacteroidetes phylum decreased close to parturition and then increased postpartum. Several members of this phylum are reported to be carbohydrate degraders, so its decrease may suggest a lower activity towards easily digestible fiber in the hindgut. Contrary to what was observed in the rumen at Pa, the relative abundance of Proteobacteria was shown to decrease.

A differential analysis of OTUs showed a significant increase in two OTUs affiliated to the Ruminococcaceae family at parturition. This family is functionally very diverse. As some *Ruminococcus* species can compete with *Fibrobacter succinogenes* for cellulose [46], the decrease in this latter genus may open ecological niches for other Ruminococcaceae family members. A differential analysis showed that the relative abundance of one OTU affiliated to the Fibrobacteres phylum decreased between BS and Pa by almost 80%. After parturition, this OTU was enriched, suggesting the quite high instability of the environment. As these results were not confirmed by qPCR, it is possible that this OTU corresponds to *Fibrobacter intestinalis*, which is the second species described for the *Fibrobacter* genus [47] and the main *Fibrobacter* species in the intestine [48], and which was not targeted by our qPCR primers. In addition, a substantial number of OTUs appeared to be impacted close to parturition, with the majority being underrepresented when compared to their relative abundance at the start of the trial (BS). This may again reflect some instability of the fecal communities in the control ewes. OTUs related to *Agaricomycetes* and *Dibaeis* fungi were identified as differential in our data (increased at Pa). These sequences probably come from some contamination from the bedding material, which may have been ingested by the animals, as at parturition the bedding could be wetter and more prone to fungal growth.

It should be noticed that high inter-individual variations were observed in the rumen while the relative abundances observed in the feces were more consistent among the animals, allowing several taxa to reach significance more easily. It can be hypothesized that as the rumen is the first digestive compartment of the digestive tract, the microbial population is more prone to be impacted by external stimuli. Parturition is known to reduce rumination behavior and thus is expected to impact the flow rate of feed, the fermentative activities, and the microbial profile of the rumen.

### 4.3. What Was the Impact of Parturition on Oxidative and Metabolic Status of the Control Ewes?

The transition period is associated with several physiological stresses. The metabolism of a late gestating ruminant is characterized by a decreased DMI and a decreased amount of glucose available in the blood, creating a negative energy balance (NEB). As already observed, during the ewes’ transition period [49], blood glucose decreased in our study as the animal reached parturition, while the NEFA concentration increased in agreement with Castillo et al. [50] in their study on dairy cows. Other liver biomarkers were not affected in our study, reflecting the correct adaptation of the gestating animal to the diet.

The transition period is also associated with an unbalanced pro-oxidant/antioxidant system producing an excess of reactive oxygen species (ROS) while the antioxidant capacity is decreased [51]. Malondialdehyde (MDA) is a marker of lipid peroxidation. MDA variations observed in the control ewes during the last month of gestation (between BS and Pa) were in line with the literature on pregnant ewes [52] and dairy cows [50], but not with other studies on ovines [49,51]. These differences are likely partly due to different methodological approaches, as in our case we precisely measured the MDA concentration using an adapted HPLC technique, whereas in the last two cited studies, MDA was only estimated through a thiobarbituric acid-reactive substances (TBARs) measurement. Only a little information exists on gestating ewes, but in dairy cows, metabolic and oxidative stresses in transition are expected to be even higher as they produce high milk quantities, thus threatening the physiological homeostasis [53]. The enzyme glutathione peroxidase (GPx) is recognized to limit lipid peroxidation. Our results showed a slight numerical increase in the GPx activity, but variations of this activity have been reported to occur depending on the physiology of the species, and even on breeds within the same species [52,53].

### 4.4. What Were the Effects of S. Cerevisiae I-1077 Supplementation around Parturition?

As expected, the *Saccharomyces* concentration increased in the rumen samples of the SC group only during the *S. cerevisiae* I-1077 (SC) supplementation period. Q-PCR data also indicated a similar increase of *Saccharomyces* at Pa in the feces of the supplemented ewes, which suggested that the SC cells were able to reach the hindgut segments during their digestive transit, which confirms earlier results [54]. The effective presence of the SC additive is important to support its action in the different compartments of the gastro-intestinal tract [55].

The SC supplementation induced some changes in the diversity and the dynamics of microbiota and fermentation, leading overall to a stabilization of the digestive conditions.

In more detail, the negative impact of parturition on *Fibrobacter* that we saw in the control group was alleviated in the SC supplemented animals in the rumen. Although our data could not reach significance, probably because of the low number of animals and inter-individual variability, we could observe different evolutions in the *F. succinogenes* concentrations between the two groups, and a stronger decrease in the relative abundance of the Fibrobacteres phylum in the control group close to parturition, which makes our hypothesis tangible. This species has shown to be quite responsive to the live yeast strain used in the present study [12,45,56]. *S. cerevisiae* I-1077 active cells may improve the rumen environment by scavenging oxygen and providing vitamins or cofactors that may help the bacterial species to maintain [12].

In the rumen of supplemented animals, a lower relative abundance of Euryarchaeota and Verrucomicrobia phyla were observed close to parturition and postpartum, respectively, compared to the control group. Euryarchaeota are methanogens in the rumen and members of the Verrucomicrobia phylum have been reported to release H_2_ from complex polysaccharide degradation [57]. The Verrucomicrobia family *RFP12* was found to be prevalent in the rumen of high methane yielding animals [58]. In a recent trial in beef cattle fed a high forage diet [59], the same strain of *S. cerevisiae* that we used also had an impact on Verrucomicrobia, with a decrease in the relative abundance of this phylum. However, further work is needed, such as methane and H_2_ quantifications, to explain the effects of SC supplementation on these taxa.

Moreover, we noticed from our 18S sequence analysis that protozoa taxonomic ruminal composition was more stable around parturition in the SC group (*Entodinium*, *Isotricha*). In addition, *Pichia* was maintained in the rumen of the SC ewes at Pa while it disappeared in the control group. It would be of interest to promote this yeast genus, as it has been recently reported that a microbial feed additive containing a strain of *Pichia* could improve the milk yield in dairy cows [60]. At Pa, the relative abundance of Ascomycota was clearly increased in the SC supplemented ewes, as expected. The beta diversity PCoA plots showed that the eukaryotic communities were different in both groups.

We noticed a more stable total VFA concentration across the periods in the SC group, whereas decreased levels of VFA were measured at parturition in the control group, mainly driven by acetate concentrations. These data suggest a beneficial effect of the live yeast supplementation on microbial activity in the rumen.

Another interesting result was the stabilization of the bacterial fecal diversity throughout the study, as indicated by the significant differences between the groups. Indeed, the supplemented animals presented stable diversity indexes over time, while a strong decrease in the alpha diversity indexes was observed in the control group at Pa. Such a sharp decrease in diversity in the hindgut may increase the risk for opportunistic pathogens to find a free ecological niche in which to settle and trigger disease [61]. Moreover, the decrease in the Bacteroidetes phylum which was observed at parturition in the control group was alleviated in the feces of the SC supplemented animals. The same observation could be raised for the Fibrobacteres phylum which was kept very stable across the whole experimental period in the SC supplemented ewes, as observed in the rumen. A stable community is generally less prone to the setup of a dysbiotic state.

Supplementation strategies to reduce oxidative stress may reduce the economic losses associated with health disorders observed during the transition period [62]. Our data suggest that supplementation by SC during late gestation alleviated oxidative stress in the ewes. Although not significant, the activity of the antioxidant enzyme GPx showed an increase of + 29% just before parturition in the SC group, while the increase was less important (+ 6%) in the control group. Moreover, the MDA level was shown to decrease in the SC group, an opposite variation to what was observed in the control animals. SC supplementation seemed to have induced pro/antioxidants production modifications. Taken together, these data are promising, but more research is needed to better understand and elucidate the molecular mechanisms by which live probiotics interact with antioxidant and immunological mechanisms. In dairy cows around calving, this live yeast has been able to modulate the inflammatory response and improve the barrier function of the rumen wall [7], with concomitant changes in the rumen microbiota diversity and composition [4]. It has been reported that rumen microbial fractions, such as bacterial LPS, would play a role in proinflammatory and prooxidative responses [8], and in periparturient dairy cattle, the concentration of lactic acid isoforms (a ruminal activity biomarker) has been correlated with the oxidant status [63]. It can thus be hypothesized that the SC effect at the rumen microbiota level could have a beneficial impact on oxidative stress. Moreover, the yeast cellular content in B vitamins could also play a role in this effect as some of these vitamins are precursors of important coenzymes involved in oxidation processes. Although we measured a higher decrease in the FRAP level at Pa in the supplemented animals, it must be reminded that the FRAP reflects only partially the antioxidant capacity as it cannot detect compounds that act by radical quenching (hydrogen transfer), particularly thiols (as glutathione), and some proteins involved in the antioxidant status of the animal [64].

Very few papers address the effect of probiotics on oxidative stress in ruminants, and even less information is available on the effect of such additives on the oxidative status of gestating ruminants. In a recent published work [65], the effect of a live yeast product was investigated in relation to oxidative stress in periparturient ewes with different impacts on the investigated parameters, but the measurements were performed 3 to 6 weeks postpartum, which may be already late regarding oxidative stress. Furthermore, the yeast strain could also be an important factor driving the effects on the oxidative status. Taken together, these data are promising but more research is needed to better understand and elucidate the molecular mechanisms by which live probiotics interact with antioxidant mechanisms.

Although not significant, one observation was a 7.2% higher birth weight on average for the lambs born from SC-supplemented ewes, compared to the controls, which would suggest a beneficial side effect of SC supplementation. We could thus hypothesize that a live yeast supply to late gestating ewes would promote nutrient transfer for fetal growth through a better-balanced digestive function and an improved oxidative status. It is known that the nutritional status of gestating dams does impact the weight of the offspring at birth, which is associated with their vigor, autonomy, and further survival capacity [66]. In particular, late-pregnancy undernutrition was associated with a significant decrease in lamb birth weight [67]. The importance of adequate maternal nutrition and the prevention of oxidative stress has been stretched, especially for ewes bearing multiple fetuses [68]. Further research is needed to confirm this benefit, by measuring the specific parameters linked to the nutritional status of the dams and to the vigor and activity of newborn offspring.

## 5. Conclusions

Overall, this study shows that the peripartum period can be a risk for digestive microbial balance and oxidative status, even in the absence of any dietary challenge. Indeed, we observed a reduced gastro-intestinal bacterial diversity and a decrease in an important plant cell wall polysaccharide-degrading bacteria and fungi due to parturition, as well as an instability in ruminal fermentative activity, which suggested that digestive homeostasis was disrupted, and we measured an increased level of oxidative stress in ewes. We showed that the supplementation of ewes during late gestation with *S. cerevisiae* I-1077 can help to maintain the key microbial communities involved in digestive efficiency and can stabilize the digestive balance throughout the gastrointestinal tract. A higher antioxidant capacity and a significant lower lipid peroxidation were also observed. In conclusion, our data suggest that live yeast prevented a transient imbalance in the homeostasis of the organism and helped the animals to cope with the stress of parturition.

## Figures and Tables

**Figure 1 jof-07-00447-f001:**
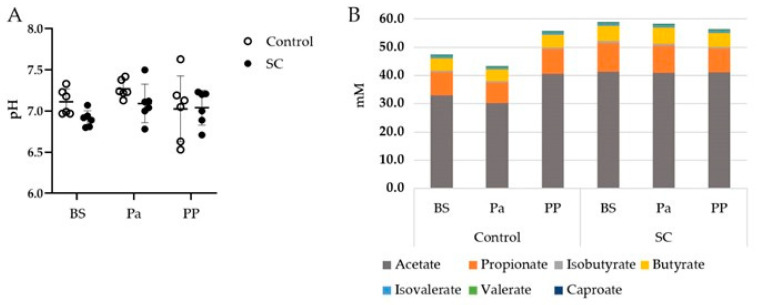
(**A**) Evolution of the rumen pH, (**B**) Mean VFA concentrations (mM) in the rumen contents of the ewes from the control or SC groups (*n* = 6 per group) over the experimental period. BS = before supplementation of SC, Pa = close to parturition, PP = 2 weeks postpartum.

**Figure 2 jof-07-00447-f002:**
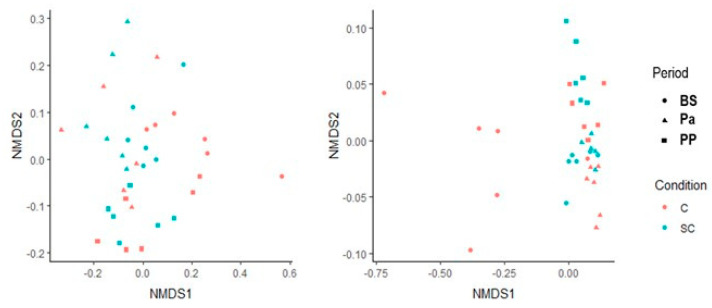
Non-metric multidimensional scaling representation for the bacterial and archaeal communities in the ruminal (**left**) and the fecal (**right**) samples. Circle = BS; triangle = Pa, square = PP.

**Figure 3 jof-07-00447-f003:**
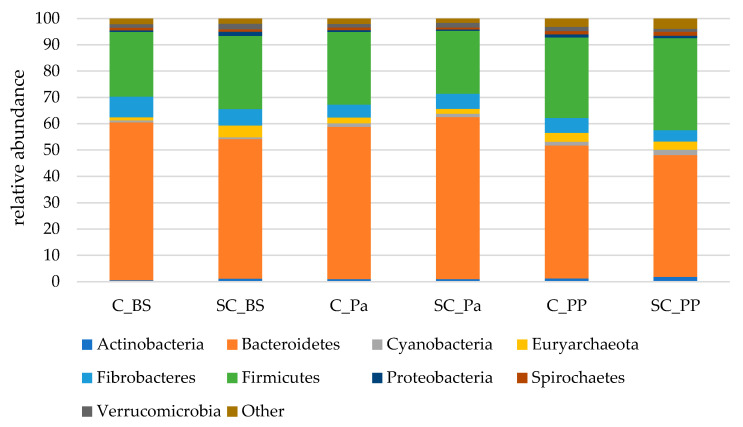
The relative abundances of the main bacterial and archaeal phyla in the rumen contents of both the experimental groups (Control C and Supplemented SC, *n* = 6 per group) BS (before supplementation), Pa (close to parturition) and PP (2 weeks postpartum).

**Figure 4 jof-07-00447-f004:**
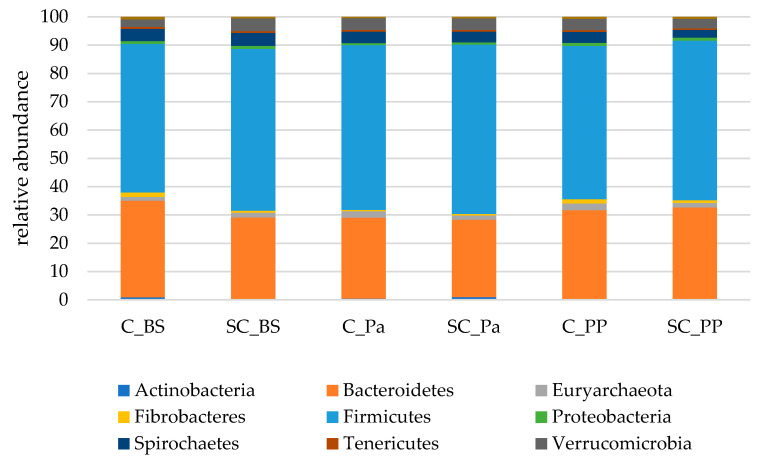
The relative abundances of the main bacterial and archaeal phyla in the fecal samples of both the experimental groups (Control C and Supplemented SC, *n* = 6 per group) BS (before supplementation), Pa (close to parturition) and PP (2 weeks postpartum).

**Figure 5 jof-07-00447-f005:**
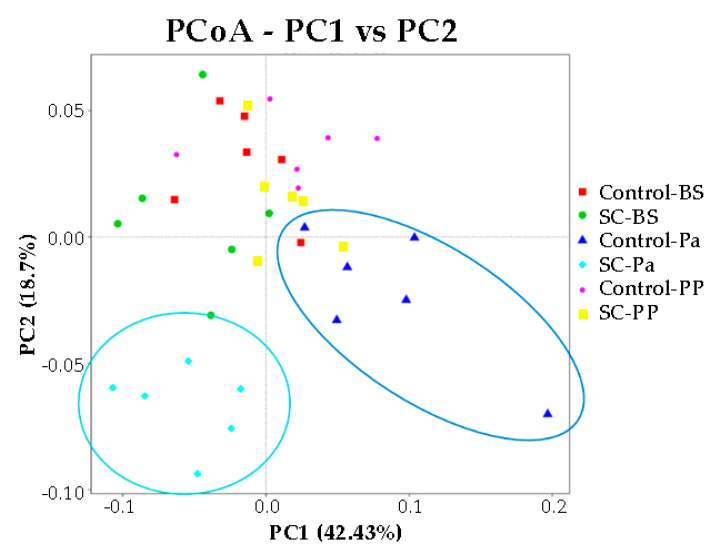
PCoA representation for the eukaryotic and fungal communities in the ruminal samples of the control and SC groups at BS, Pa, and PP.

**Figure 6 jof-07-00447-f006:**
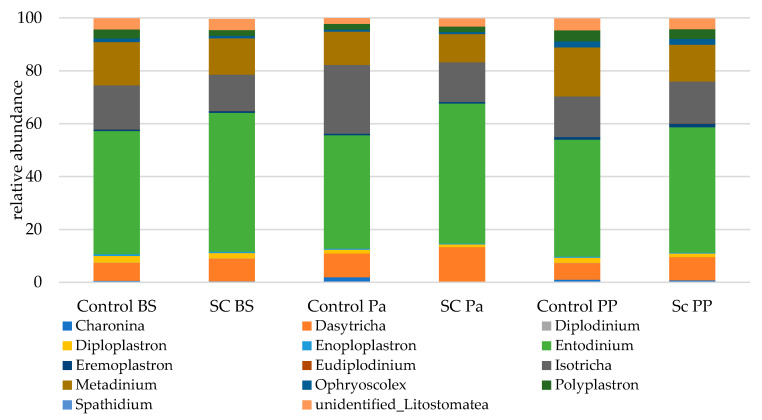
Relative abundance (% eukaryotic population) of protozoa in the ruminal samples from the control and SC groups (*n* = 6 per group) at BS, Pa and PP.

**Figure 7 jof-07-00447-f007:**
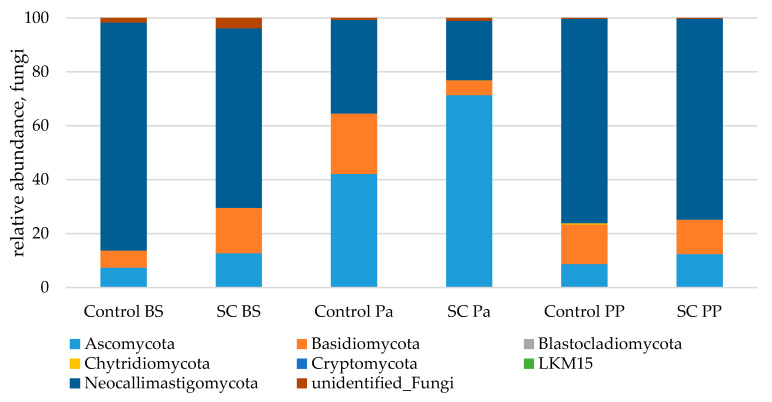
Relative abundance (%) of fungi in the ruminal samples from the control and SC groups (*n* = 6 per group) at BS, Pa and PP.

**Figure 8 jof-07-00447-f008:**
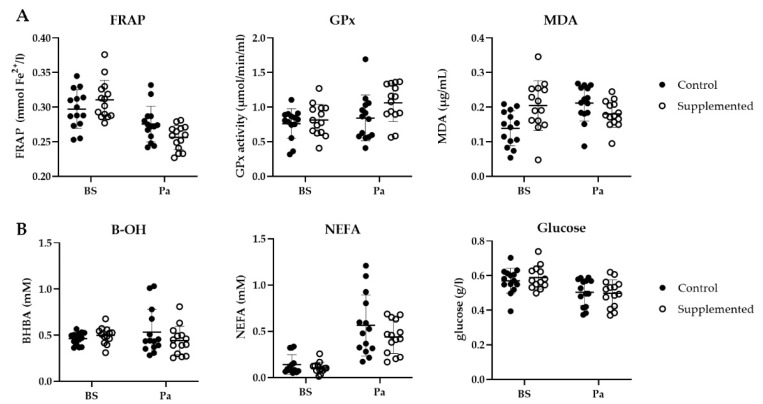
(**A**) Ferric reducing ability of plasma (FRAP, mmol Fe^2+^/L), GPx activity (µmol/min/mL) and plasmatic MDA concentration (µg/mg protein); (**B**) BHBA (mM), NEFA (mM) and glucose (g/L) in the control (full circle) or the SC (open circle) ewes before supplementation (BS) and around parturition (Pa) (*n* = 14 per group).

**Table 1 jof-07-00447-t001:** *p*-values associated with the VFA statistical analysis with a linear mixed model.

Measured Parameter	Group	Time	Interaction G × T
Total VFA	0.309	0.164	0.060
Acetate	0.357	0.080	0.086
Propionate	0.131	0.722	0.047
Butyrate	0.569	0.647	0.055
Isobutyrate	0.891	0.168	0.136
Isovalerate	0.870	0.607	0.131
Valerate	0.535	0.002	0.085
Caproate	0.142	0.002	0.180

**Table 2 jof-07-00447-t002:** Q-PCR microbial quantification results for the targeted microbial groups or species in the ruminal samples (*n* = 6 per group) and the *p*-values associated with the statistical analysis performed on Y_diff_ with a linear mixed model. BS = before supplementation of SC, Pa = close to parturition, PP = 2 weeks postpartum. NA indicates that the linear mixed model was not applied to these data.

Log_10_ of Copy Numbers/g Pelleted Rumen Content	BS		Pa		PP		*p*-Value from Linear Mixed Model on Y_diff_		
Target	Control	SC	Control	SC	Control	SC	Group	Time	Interaction G X T
Total bacteria	10.86 ± 0.09	10.83 ± 0.18	10.91 ± 0.11	10.91 ± 0.24	10.78 ± 0.07	10.78 ± 0.17	0.648	0.019	0.941
*Ruminococcus flavefaciens*	7.72 ± 0.32	8.09 ± 0.42	8.03 ± 0.16	7.79 ± 0.29	7.8 ± 0.12	7.8 ± 0.23	0.013	0.181	0.152
*Ruminococcus albus*	6.64 ± 0.16	6.9 ± 0.24	6.85 ± 0.35	6.99 ± 0.47	7.16 ± 0.25	7.19 ± 0.22	0.157	0.003	0.403
*Fibrobacter succinogenes*	9.61 ± 0.17	9.37 ± 0.38	9.28 ± 0.10	9.32 ± 0.19	9.32 ± 0.38	9.24 ± 0.16	NA	NA	NA
*Prevotella*	8.93 ± 0.12	8.94 ± 0.19	8.96 ± 0.11	9.01 ± 0.16	8.84 ± 0.17	8.76 ± 0.18	0.831	0.013	0.315
*Megasphaera elsdenii*	2.48 ± 0.32	2.42 ± 0.17	2.41 ± 0.5	2.68 ± 0.17	2.55 ± 0.32	2.70 ± 0.31	0.438	0.234	0.825
Methanogenic Archaea	7.66 ± 0.23	7.87 ± 0.23	7.95 ± 0.29	8.07 ± 0.19	7.85 ± 0.41	8.02 ± 0.21	0.726	0.129	0.923
Protozoa	9.84 ± 0.14	9.52 ± 0.66	9.92 ± 0.34	9.33 ± 0.52	9.51 ± 0.15	9.28 ± 0.31	0.786	0.106	0.200
Anaerobic fungi	6.27 ± 0.37	6.19 ± 0.64	6.36 ± 0.71	6.13 ± 0.49	5.94 ± 0.72	6.16 ± 0.39	0.819	0.486	0.426
*S. cerevisiae*	5.89 ± 0.49	5.65 ± 0.38	6.12 ± 0.6	7.49 ± 0.34	5.6 ± 0.18	5.91 ± 0.54	0.027	0.001	0.003

**Table 3 jof-07-00447-t003:** *p*-values associated with the statistical analysis of the alpha diversity indexes of the bacterial and archeal communities in the ruminal samples with a linear mixed model.

Alpha-Div Indexes	Group	Time	G × T
Observed OTUs	0.589	0.001	0.626
Shannon	0.851	0.003	0.178

**Table 4 jof-07-00447-t004:** *p*-values associated with the statistical analysis of the alpha diversity indexes of the bacterial and archeal communities in the fecal samples with a linear mixed model.

Alpha-Div Indexes	Group	Time	G × T
Observed OTUs	0.017	0.010	0.335
Shannon	0.013	0.264	0.229

**Table 5 jof-07-00447-t005:** *p*-values associated with the statistical analysis of the relative abundances of the bacterial and archaeal communities in the ruminal samples with a linear mixed model.

Phylum	Group	Time	G × T
Actinobacteria	0.589	0.039	0.503
Bacteroidetes	0.225	0.044	0.445
Cyanobacteria	0.254	0.138	0.312
Euryarchaeota	0.051	0.260	0.952
Fibrobacteres	0.565	0.747	0.241
Firmicutes	0.562	0.138	0.371
Proteobacteria	0.259	0.006	0.776
Spirochaetes	0.672	0.013	0.219

**Table 6 jof-07-00447-t006:** *p*-values associated with the statistical analysis of the relative abundances of the bacterial and archaeal communities in the fecal samples with a linear mixed model.

Phylum	Group	Time	G × T
Bacteroidetes	0.044	0.034	0.523
Euryarchaeota	0.355	0.928	0.920
Fibrobacteres	0.066	0.0001	0.013
Firmicutes	0.442	0.146	0.891
Proteobacteria	0.614	0.004	0.936
Spirochaetes	0.241	0.414	0.495
Tenericutes	0.914	0.357	0.953
Verrucomicrobia	0.183	0.453	0.712

**Table 7 jof-07-00447-t007:** *p*-values associated with the statistical analysis of the relative abundances of eukaryote phyla in the ruminal samples with a linear mixed model. Only genera with relative abundance >2.3% were kept for statistical analysis, except *Eudiplodinium* whose abundance was <2.3% but which is a functionally important ciliate genus in the rumen.

Protozoa	Group	Time	G × T
*Dasytricha*	0.224	0.007	0.514
*Entodinium*	0.843	0.291	0.085
*Eudiplodinium*	0.871	0.899	0.041
*Isotricha*	0.625	0.032	0.015
*Metadinium*	0.799	0.0002	0.115
*Polyplastron*	0.14	0.001	0.550
Unidentified Litostomatea	0.811	0.013	0.284
Total Litostomatea	0.150	0.302	0.445

**Table 8 jof-07-00447-t008:** *p*-values associated with the statistical analysis of the relative abundances of fungi phyla in the rumen. Only phyla with a relative abundance >1% were kept for statistical analysis.

Phylum	Group	Time	G × T
Ascomycota	0.103	<0.0001	0.065
Basidiomycota	0.009	0.971	0.289
Neocallimastigomycota	0.229	<0.0001	0.528
Unidentified fungi	0.209	0.001	0.111

**Table 9 jof-07-00447-t009:** *p*-values associated with the statistical analysis of the relative abundances of anaerobic fungi phyla in the ruminal samples with a linear mixed model.

Anaerobic Fungi	Group	Time	G × T
*Cyllamyces*	0.500	0.007	0.179
*Orpinomyces*	0.825	0.201	0.704
Unidentified Neocallimastigomycota	0.180	<0.0001	0.604
Total	0.236	<0.0001	0.543

## Data Availability

Sequencing data are available in the BioProject SRA database (https://www.ncbi.nlm.nih.gov/sra/, accessed on 3 June 2021) as PRJNA713537.

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
