# Peer review of "Changes in Digestive Microbiota, Rumen Fermentations and Oxidative Stress around Parturition Are Alleviated by Live Yeast Feed Supplementation to Gestating Ewes"

_jof, 2021, doi:10.3390/jof7060447_

Round 1

Reviewer 1 Report

After reading and reviewing the manuscript, these are my comments:

- Line 89-90: the authors do not report how the additive is included with the feed. They also do not report on whether they have monitored individual feed intake to ensure that animals ingest the dose of additive. 

- Line 101-102. The authors do not report on the nutritional composition of the feed and forage or the UFL they provide. Nor do they report on the microbiological quality of the feed, when the additive used is a yeast.

- Line 113-114: What is the method used to identify the CFU?

- Line 134: The quality of the rumen sample is visually checked (absence of saliva), but in the discussion it is indicated that the results obtained for pH of the rumen contents may be influenced by the presence of saliva.

- Line 151: What fraction of rumen have you processed? The solid fraction? Has it been previously filtered?

- Line: 162-165: It would be advisable to expand the description of the methodology used to prepare the standards for each target group. According to the bibliography provided, it would be explained for cellulolytic bacteria and SC.

- Line 179-180: What is the criteria used to select 6 animals out of the 14 available in each group for metagenomics studies?

- Line 203; protocols, respectively. 

- Line 190-205: indicate whether commercial kits have been used (based on the literature indicated).

- Line 184-186: further information on the software used in the bioinformatics analysis (QIIME, Mothur, etc.) is recommended.

- Line 207-213: specify the methods used: kinetic, colorimetric, etc.

- Line 216-231: I have not been able to find the information in section 2-1 to evaluate the zootechnical parameters of the animals. There is contradictory information on how the animals are assigned to each experimental group. There is also a lack of information on the programme used to perform the statistical analyses. It should be reviewed whether the normalisation (subtraction of the initial BS values) is adequate, or whether these data should be included as covariates in the statistical model used.

- Results: In general, results and discussion are presented in this section. In the opinion of the reviewer, the tables and graphs presented do not give precise information on the significant differences found between the experimental groups. It is difficult to make an assessment of the significant differences between the two experimental groups. As an example, Table 2 and Table 3 and R. flavefaciens. The authors describe many results, and increases and decreases in the percentage of bacterial groups, but without the support of statistical significance. Other results that differ statistically refer to the sampling time (probably associated with the physiological state of the animal) and not to the experimental treatments studied, which is the objective of the study.

- Discussion: In general, the discussion is very extensive and not very targeted to the objective of the study. At times, the discussion seems more like a comparison between molecular techniques (qPCR vs. 16/18 rDNA). The discussion focuses too much on the control group of animals and furthermore, it focuses on results that are not statistically significant.

- Specifically, in lines 677-679, contamination of faecal samples by bedding material is argued, when the materials indicate that faeces are taken directly from the rectum.  In lines 762-768, results related to methane production are argued, which are not accompanied by significant changes in AFV.

- The conclusion is very general and not oriented to the results obtained, in particular concerning the efficiency and digestive stabilisation of the microbiological communities of the gastrointestinal tract.

Author Response

- Line 89-90: the authors do not report how the additive is included with the feed. They also do not report on whether they have monitored individual feed intake to ensure that animals ingest the dose of additive

Answer: The yeast additive is included during the pelleting process. We used a coated form of the additive (which is commercialized as TITAN form of Levucell SC), which protects the yeast cells from degradation during the pelletization process. This information has been included in the text, lines 93-95 of the revised version.

We did control the concentrate intake at the individual level using the Biocontrol CFRI, as reported in the manuscript (lines 76-78 of the revised version). However, it was maybe not clear enough, so we modified the sentence to improve understanding, lines 78-81 of the revised version.

- Line 101-102. The authors do not report on the nutritional composition of the feed and forage or the UFL they provide. Nor do they report on the microbiological quality of the feed, when the additive used is a yeast.

Answer: The composition of the concentrate was given in Table S1 in the supplementary material. In the revised version, we added a Table S2 in which we also give the nutritional composition of the diet as requested, and to which we refer to on line 88 of the revised version.

There was no particular control of the microbial quality of the feed, except for yeast detection in the concentrate, as reported. Concentrates are generally very dry (~85% of DM) and due to their physical form of presentation (pellets) and their production process, they are generally very low in microbial contaminations. The meadow hay was of good quality, and a microbiological control was not necessary as no visible sign of spoilage mould or any particular microbial contamination were noticed. However, this is true that forage is not sterile and possibly contains wild yeasts that may enter the rumen, and this would have been the case for all ewes from both groups. Indeed, as shown on figure 2, we detected some S. cerevisiae DNA in the rumen and feces of the two groups before the distribution of SC, and a significant increase in S. cerevisiae abundance was measured only in the SC group when the feed additive had been distributed.

- Line 113-114: What is the method used to identify the CFU?

Answer: We did not identify the CFU that were counted on the agar plates. In fact, we also checked the Control concentrate, and it was devoid of any cultivable yeast (no colony appeared on Control plates). The colonies that grown on the SC-supplemented concentrate were visually not moulds, and their aspect was similar to that generally obtained with Saccharomyces cerevisiae, so we considered that they were most probably of the SC yeast additive. It is true that we did not report in the manuscript that we also checked the Control concentrate, so we added this information for clarification, lines 115-116 of the revised version.

- Line 134: The quality of the rumen sample is visually checked (absence of saliva), but in the discussion it is indicated that the results obtained for pH of the rumen contents may be influenced by the presence of saliva.

Answer: Yes, you are right, we just visually checked for absence of visible amount of saliva, but contamination may still occur with small amounts which cannot be detected visually. So, we could not guarantee that all samples were not contaminated, and as pH of samples were close to neutrality or a bit higher than 7.0, we could not exclude that minor contamination occurred and induced a bias in our pH measurements. A precision was added in the materials and methods section to indicate that we checked for the absence of visible amount of saliva, line 139 of the revised version.

- Line 151: What fraction of rumen have you processed? The solid fraction? Has it been previously filtered?

Answer: The rumen fluid was collected with a stomach tube, so we obtained quite liquid samples. The samples were not filtered but prepared following different protocols according to the measured parameters as reported in 2-3. The use of unfiltered rumen fluid after sampling by intubation has been published previously (eg, in Chaucheyras-Durand et al. 2019, Scientific reports, doi:10.1038/s41598-019-55825-0; Moon et al. 2021, doi: 10.1038/s41598-021-82815-y)

- Line: 162-165: It would be advisable to expand the description of the methodology used to prepare the standards for each target group. According to the bibliography provided, it would be explained for cellulolytic bacteria and SC.

Answer: In the first version of our manuscript, we had decided not to describe in detail the protocols for standards preparation to avoid too long material and methods section. Indeed, these protocols have been described in the cited publications. We just reported with little more detail what we did for SC, because the protocol was slightly different from the other targets where we used a range of target gene copies/g of sample (and not cells/g). However, as requested we provide additional information and we included a column in Supplementary Table S3 reporting the microbial material used to generate standard curves.

- Line 179-180: What is the criteria used to select 6 animals out of the 14 available in each group for metagenomics studies?

Answer: As DNA sequencing studies are still quite expensive, we could unfortunately not analyze all the samples. Animals were randomly chosen in the two groups among those with complete set of samples.

- Line 203; protocols, respectively. 

Answer: Modification has been made. Thanks.

- Line 190-205: indicate whether commercial kits have been used (based on the literature indicated).

Answer: The literature in this section for liver and blood metabolites measurements are not referring to commercial kits that is the reason why we didn’t indicate any commercial reference.

- Line 184-186: further information on the software used in the bioinformatics analysis (QIIME, Mothur, etc.) is recommended.

Answer: Bioinformatics analysis was performed by sequencing company Novogene according to their internal pipeline. Description of each bioinformatic step, with appropriate references, has been added in the text, lines 186-193 of the revised version.

- Line 207-213: specify the methods used: kinetic, colorimetric, etc.

Answer: The methods cited were using spectrophotometry for measurement of liver and blood metabolites with only MDA using UV-Spectrophotometry as indicated lines 201, 202, 205, 207 of the revised version.

- Line 216-231: I have not been able to find the information in section 2-1 to evaluate the zootechnical parameters of the animals. There is contradictory information on how the animals are assigned to each experimental group. There is also a lack of information on the programme used to perform the statistical analyses. It should be reviewed whether the normalisation (subtraction of the initial BS values) is adequate, or whether these data should be included as covariates in the statistical model used.

Answer: As reported in 2-1, the group assignment was determined according to age, parity, body condition score, and live weight of ewes, and we totally agree that these parameters could not be properly defined as zootechnical parameters. Thus, the text was modified to take this comment into account, line 217 of the revised version.

The program used for statistical analysis was the same as the one used for graphics representation. We added this information in the text (line 215 of the revised version). We added also more explanation and references to clarify and support the statistical method we used for analyzing our data, lines 218-224 of the revised version.

- Results: In general, results and discussion are presented in this section. In the opinion of the reviewer, the tables and graphs presented do not give precise information on the significant differences found between the experimental groups. It is difficult to make an assessment of the significant differences between the two experimental groups. As an example, Table 2 and Table 3 and R. flavefaciens. The authors describe many results and increases and decreases in the percentage of bacterial groups, but without the support of statistical significance. Other results that differ statistically refer to the sampling time (probably associated with the physiological state of the animal) and not to the experimental treatments studied, which is the objective of the study.

Answer: In fact, the tables present the statistical results from the normalized data while the graphs show the data collected from each time point, including BS. Thus, from our point of view it is not possible to show statistical results (e.g., p-values represented with letters) of group or time effects on the graphs, as those reflect the evolution of each parameter across periods. We thought that tables and graphs would be rather complementary from each other. However, we fully understand the comment and we are aware that this may be a little bit confusing. To improve reading easiness, we propose to include, in the supplementary material, new graphs representing the “normalized” data, now named Ydiff , on which statistical results are reported. We really hope that this additional material will be helpful to improve our manuscript.

However, due to this new presentation, the number of Supplementary Figures have been greatly increased. In the case of 14 supplementary figures represent a too large number of added materials, we could suggest deleting the 5 Supplementary Figures depicting the DESeq2 analysis as the results are presented in the text and discussed, and the graphical representations might not be mandatory to understand the biological meaning of these results. Similarly, we have deleted the last Supplementary Figure of the ASL, ALT, PAL, and GGT values where no significant difference was observed according to Time, Group or interaction.

- Discussion: In general, the discussion is very extensive and not very targeted to the objective of the study. At times, the discussion seems more like a comparison between molecular techniques (qPCR vs. 16/18 rDNA). The discussion focuses too much on the control group of animals and furthermore, it focuses on results that are not statistically significant.

Answer: We thank you very much for this comment. In fact, we had two objectives in this work: i) better characterize the peripartum period and the potential changes in gastro-intestinal microbiota, fermentations, and oxidative stress, ii) investigate the impact of live yeast supplementation. However, from your comment it clearly appears that we need to re-balance our discussion. Therefore, we propose to slightly change the title of our manuscript in order to get a better adequation with our objectives, and we made modifications in the abstract, introduction and discussion accordingly (lines 13-16, 17-19, 23-24, 61-65, and suppression of the first sentence of the discussion in the revised version). Our aim was not to compare molecular techniques per se but to use those to generate complementary data, as qPCR gives quantitative assessment of selected microorganisms, whereas DNA sequencing provides information on relative abundance of taxa, diversity, and more global composition of the microbiota. However, we understand your comment and we made changes in the discussion to avoid comparisons of techniques, lines 596, 601, 606, 663 of the revised version.

- Specifically, in lines 677-679, contamination of faecal samples by bedding material is argued, when the materials indicate that faeces are taken directly from the rectum.  In lines 762-768, results related to methane production are argued, which are not accompanied by significant changes in AFV.

Answer: We understand your comment. Indeed, our sentence was not clear. Fecal material has been collected directly from the rectum and thus cannot be directly contaminated with bedding material. Nevertheless, animals may ingest bedding material, which can lead to transient contamination of the gastro-intestinal tract with the mentioned taxa. We slightly modified the sentence for better clarity, lines 685-687 in the revised version, and we reduced the discussion around the fungal sequences probably coming from consumption of contaminated material.

Regarding the hypothesis on methane, we measured VFA concentration, which is the result of production and absorption, which makes actually very difficult to interpret VFA data. However, we fully agree with your comment and we modified the sentence accordingly (lines 766-768 in the revised version).

- The conclusion is very general and not oriented to the results obtained, in particular concerning the efficiency and digestive stabilisation of the microbiological communities of the gastrointestinal tract.

Answer: we agree with your comment, we made modifications to remind the most important findings of our work to avoid too general conclusion, lines 846-856 of the revised version. We also moved the sentence associated with dairy cows to the discussion (see lines 581-584 of the revised version).

Reviewer 2 Report

The study is novel and interesting, as it is discussed the beneficial effect of live yeast supplementation in prepartum ewes on their rumen and gut microbiota balance, fermentative activities, as well as, their redox and metabolic status.

However, the study is very analytical, especially in Materials and Methods (e.g. lines 140-148) and Results, where the information could be less. Overall, the manuscript is too extended and a bit laborious to comprehend it.

Some minor revisions are recommended as detailed below:

Lines 98-99: please delete the sentence, it is a useless repetition.

Line 215: should be written in 2.1

In figure 9, please write the parameters in the legend in the order that are presented in the graphs.

Lines 539-541: about the remaining ewes that did not give birth to twin lambs, how many lambs did they give birth, more, less or none? It must be written the number of born lambs per ewe. Otherwise, the reader might understand that a false foetal count was performed.

Line 803: ‘to an overall better health status of the ewe around parturition?’ This statement is non-objective, as neither Control nor SC ewes showed any health disorder. ‘Taken together, these data are promising but more research is needed to better understand and elucidate the molecular mechanisms by which live probiotics interact with antioxidant and immunological mechanisms’

Lines 539-543 vs 834-845: It is written that the birth weight of neonate lambs was higher in those born from SC ewes compared with those from Control ewes. This is not true as no significant statistical difference was found (539-546). The paragraph and the statements, concerning the birth weight of newborn lambs, should be rephrased.

Lines 854-855: the words ‘beneficial effects on ewes’ health around parturition should be revised appropriately, because any alteration in redox status, especially around parturition, does not indicate worst health condition but a transient imbalance in homeostasis of the organism. Thus, the live yeast supplementation could be beneficial for the periparturient ewes to cope with the stress of parturition.

Author Response

The study is novel and interesting, as it is discussed the beneficial effect of live yeast supplementation in prepartum ewes on their rumen and gut microbiota balance, fermentative activities, as well as their redox and metabolic status.

Answer: We thank you very much for your comment and we are happy you appreciated our manuscript.

However, the study is very analytical, especially in Materials and Methods (e.g. lines 140-148) and Results, where the information could be less. Overall, the manuscript is too extended and a bit laborious to comprehend it.

Answer: We understand your comment. We tried to reduce some paragraphs in order to improve the easiness of reading. Please see the modifications in the revised version (e.g., in the Material and methods section, we have reduced the paragraph concerning blood sample collection; we have reduced the paragraph on oxidative status parameters measurement; in the Results section, we have reduced the paragraph on rumen VFA, on fecal qPCR, and at several levels in the Discussion section).

Some minor revisions are recommended as detailed below:

Lines 98-99: please delete the sentence, it is a useless repetition.

Answer: We deleted the sentence as recommended.

Line 215: should be written in 2.1

Answer: you are right. We moved this sentence in the 2.1 section as recommended.

In figure 9, please write the parameters in the legend in the order that are presented in the graphs.

Answer: This has been corrected.

Lines 539-541: about the remaining ewes that did not give birth to twin lambs, how many lambs did they give birth, more, less or none? It must be written the number of born lambs per ewe. Otherwise, the reader might understand that a false foetal count was performed.

Answer: the ewes that did not give birth to twins had either one single, or three lambs. We added this information in the text, lines 563-565 of the revised version.

Line 803: ‘to an overall better health status of the ewe around parturition?’ This statement is non-objective, as neither Control nor SC ewes showed any health disorder. ‘Taken together, these data are promising but more research is needed to better understand and elucidate the molecular mechanisms by which live probiotics interact with antioxidant and immunological mechanisms’

Answer: We fully agree with your comment and we have removed this statement. We thank you for your suggestion of rephrasing, that we have included in the revised version, lines 801-803.

Lines 539-543 vs 834-845: It is written that the birth weight of neonate lambs was higher in those born from SC ewes compared with those from Control ewes. This is not true as no significant statistical difference was found (539-546). The paragraph and the statements, concerning the birth weight of newborn lambs, should be rephrased.

Answer: we acknowledge that the differences in birth weight between the two groups of lambs was not significant, however we observed a 7.2% higher weight in average in lambs born from SC ewes, which represents almost 300 g, and we considered this numerical difference quite interesting to report given the importance of this parameter in the lamb vitality and survival rate. Therefore, we modified the sentence to both fit with your recommendations and keep this observation in the discussion. Please see the rephrased sentence, lines 832-834 of the revised version.

Lines 854-855: the words ‘beneficial effects on ewes’ health around parturition should be revised appropriately, because any alteration in redox status, especially around parturition, does not indicate worst health condition but a transient imbalance in homeostasis of the organism. Thus, the live yeast supplementation could be beneficial for the periparturient ewes to cope with the stress of parturition.

Answer: we understand your comment and agree that we should not state that health is improved. Your suggestion of rephrasing is perfect to us; thus, we have included the modifications accordingly, lines 853-856 of the revised version. Thanks for your input.

Round 2

Reviewer 1 Report

The authors have made a considerable effort to improve the article and make it easier to understand. However, some additional issues remain:

 Abstract: Line 18: ruminal

Table S2: Thank you for inclusion. Information on the fat and protein values of the diet components is missing, as in the opinion of the reviewer, the nutritional composition of the diet has an important influence on the rumen and faecal microbial populations.

L180: What criteria were used for the selection of the subgroup of 6 animals for the metagenomics studies? Randomly? The groups remained balanced after this selection?

Line 186-192. Thank you for extending the information on the bioanalysis, but I think it is not complete, as the information of the programme used for the OTU abundance differentiation analyses is missing.

2.4.- Statistical analyses. Thank you for expanding the description and providing the reference to the "anova of change" model.  However, it is not clear which parameters showed significant differences in the BS period. It is advisable to define the parameters. In the reviewer's opinion the statistical model of the study is a 2 x 3 factorial, i.e. two treatments and in three times, even if it is not supplemented at one time.  Running this model would make it easier to present the results in a single table (e.g. to merge the data from tables 2, 4 and 4 into a single table) and to better interpret the differences. It would also make it easier to indicate for each variable the statistical method applied (e.g. with subscripts).

3.- Results. The new material provided as supplementary figures facilitates the interpretation of the results in a better way than the figures and tables included in the paper itself, especially when the new variable Ydiff has been described. Consider replacement.

4.-DIscussion. Some changes have been made, but I think it is still very extensive, and it would be advisable to focus the discussion on relevant results or on those having biological value.

Author Response

The authors have made a considerable effort to improve the article and make it easier to understand. However, some additional issues remain:

Abstract: Line 18: ruminal

Modification has been made, thanks.

Table S2: Thank you for inclusion. Information on the fat and protein values of the diet components is missing, as in the opinion of the reviewer, the nutritional composition of the diet has an important influence on the rumen and faecal microbial populations.

This information has been included, thank you for this comment.

L180: What criteria were used for the selection of the subgroup of 6 animals for the metagenomics studies? Randomly? The groups remained balanced after this selection?

We took a great care in selecting samples and the final sub-groups were balanced in terms of age, weight and BSC. This information was added in the text lines 181-182 of the revised version.

Line 186-192. Thank you for extending the information on the bioanalysis, but I think it is not complete, as the information of the programme used for the OTU abundance differentiation analyses is missing.

You are correct, the information was missing and was added lines 193-194 of the revised version. Thanks.

2.4.- Statistical analyses. Thank you for expanding the description and providing the reference to the "anova of change" model.  However, it is not clear which parameters showed significant differences in the BS period. It is advisable to define the parameters. In the reviewer's opinion the statistical model of the study is a 2 x 3 factorial, i.e. two treatments and in three times, even if it is not supplemented at one time.  Running this model would make it easier to present the results in a single table (e.g. to merge the data from tables 2, 4 and 4 into a single table) and to better interpret the differences. It would also make it easier to indicate for each variable the statistical method applied (e.g. with subscripts).

We thank the reviewer for his/her comments and recommendations.

The list of the parameters showing statistical difference or tendency (p < 0.1) at BS was provided as required in the statistical analysis section lines 219-222 of the revised version.

We agree with the reviewer on the 2 x 3 factorial model of our study. Initially we applied a 2*2 model as BS diff was null for each parameter, but the statistical power of the 2 x 3 model is indeed better, so, as recommended by the reviewer, we moved to this approach for the revised version. It led to a better QQplot for several items which were not tested anymore through non parametric Mann Whitney test (valerate, caproate for VFA concentrations and methanogenic archaea, S.cerevisiae and R. albus for qPCR data). Each modification has been highlighted in yellow in the text all along the Results section. For all the other items initially tested through mixed model, the new p-values were similar to those obtained with the 2 x 2 model thus our previous interpretation was unchanged. The table 2 presenting measured values at BS, Pa and PP was kept but we added columns reporting the p-values calculated with this new 2 x 3 model, with the precision that the model was applied on Ydiff, and not on the initial BS, Pa and PP values (Table 2 of the revised version). It was still impossible to add subscripts as recommended as the presented p-values correspond to Ydiff but we thought of a higher biological interest to present BS, Pa and PP values. Thus Table 3 and 4 of the initial version were removed and the following tables were re-numbered accordingly in the revised version. Finally, S. cerevisiae qPCR data in feces were tested through linear mixed model and these results are now presented in a new Supplementary Figure S3. We renumbered the supplementary figures accordingly.

3.- Results. The new material provided as supplementary figures facilitates the interpretation of the results in a better way than the figures and tables included in the paper itself, especially when the new variable Ydiff has been described. Consider replacement.

We thank very much the reviewer for his/her comment. However, the authors consider the biological “raw” data measured at BS, Pa and PP of high interest for literature comparison and thus propose to keep the main Figures unchanged in the text while presenting Ydiff Figures for each significantly affected parameter in the Supplementary material. It also allows to restrain the number of panels presented in each Supplementary Figure as only the significantly affected ones are shown.

4.-DIscussion. Some changes have been made, but I think it is still very extensive, and it would be advisable to focus the discussion on relevant results or on those having biological value.

We agree with this/her comment and several additional modifications have been made along the discussion section to reduce it by focusing on relevant results (Lines 682-685; 691-694; 697-701; 774-780 of the revised version).

Round 3

Reviewer 1 Report

Thank you for the work done to improve the manuscrit and the presentation of the results. 
On the other hand, I think the discussion is still extensive. 

In the abstact, consider referring to the terms rumen and gut instead of gastrointestinal microbiota.